# HIV incidence among sexually active young males and females in Kisumu County, Western Kenya

John Owuoth[1,2], Chiaka Nwoga[3,4], Valentine Sing'oei[1,2], Adam Yates[3,4], June Otieno[2], Erica Broach[3,4], Qun Li[3,4], Zebiba Hassen[3,4], Seth Frndak[3,4], Michelle Imbach[3,4], Leigh Anne Eller[3,4], Mark Milazzo[3,4], Tsedal Mebrahtu[3,4], Hunter J. Smith[2], Nathanial K. Copeland[2], Jessica Cowden[2], Dale J. Hu[5], Merlin L. Robb[4], Julie A. Ake[3], Christina S. Polyak[3,4], Trevor A. Crowell[3,4]*, on behalf of the RV393 Study Group¶

**1** HJF Medical Research International, Kisumu, Kenya, **2** Walter Reed Army Institute of Research-Africa, Nairobi, Kenya, **3** U.S. Military HIV Research Program, CIDR, Walter Reed Army Institute of Research, Silver Spring, Maryland, United States of America, **4** Henry M. Jackson Foundation for the Advancement of Military Medicine, Bethesda, Maryland, United States of America, **5** Division of AIDS, National Institute of Allergy and Infectious Disease, National Institutes of Health, Bethesda, Maryland, United States of America

¶ Complete membership of the RV393 Study Group can be found in the Acknowledgments
* tcrowell@hivresearch.org

## Abstract

### Introduction

HIV prevalence in Kisumu County, Kenya, is over four times the national average. We estimated HIV incidence among adults with multiple sexual partners and identified risk factors associated with HIV acquisition.

### Methods

We enrolled adults aged 18–35 years reporting ≥2 sexual partners in the prior 3 months, recruiting primarily from villages with fisherfolk. HIV counseling and testing were performed every 3 months for up to 24 months. Demographic, health, sexual, and behavior questionnaires were administered every 6 months. Enrollment characteristics were compared between participants who did and did not acquire HIV using chi-squared tests. Bivariable Cox proportional hazards models estimated hazard ratios (HRs) and 95% confidence intervals (CIs) for potential risk factors.

### Results

Between January 2017 and August 2021, 619 participants without HIV were enrolled, of whom 45% were female, 47% were ≥25 years old, 66% were single or widowed, and 46% had some secondary education. Eleven HIV seroconversions occurred over 1117.9 person-years (PY; incidence 9.84/1000 PY [95% CI: 5.22–17.03]), including eight in the first year of follow-up (13.84/1000 PY). Risk factors in unadjusted

**Data availability statement:** The minimal data set is available at Harvard Dataverse via https://dataverse.harvard.edu/dataset.xhtml?persisten-tId=doi:10.7910/DVN/1KECKH. For questions, please contact the Data Coordinating and Analysis Center (DCAC) at PubRequest@hivre-search.org and indicate the RV393 study along with the name of the manuscript.

**Funding:** This work was supported by a cooperative agreement (W81XWH-11-2-0174; W81XWH-18-2-0040) between the Henry M. Jackson Foundation for the Advancement of Military Medicine, Inc., and the U.S. Department of Defense. This research was funded, in part, by the U.S. National Institute of Allergy and Infectious Diseases.

**Competing interests:** The authors report no potential competing interests.

**Abbreviations:** CI, confidence interval; HR, hazard ratio; IQR, interquartile range; IR, incidence rate; KEMRI, Kenya Medical Research Institute; MSM, men who have sex with men; PY, person years; SES, socioeconomic status; STI, sexually transmitted infections; WRAIR, Walter Reed Army Institute of Research.

analyses included ≥4 sexual partners (HR 4.39 [95% CI 1.10-17.55]) and forced sexual activity (HR 9.15 [95% CI 1.17-71.51]).

## Conclusion

HIV incidence was lower than expected, possibly reflecting prevention efforts or COVID-19-related behavior changes. Lower incidence in later follow-up may indicate a cohort effect from regular counseling. Increasing ART coverage and pre-exposure prophylaxis may have decreased HIV incidence in fishing communities of Western Kenya.

## Background

An estimated 1.5 million people acquired HIV worldwide in 2021, with 51% of all new cases globally occurring in sub-Saharan Africa. Though this region bears a disproportionate burden of HIV compared to other parts of the world, there has been a regional decline in HIV incidence since 2010 [1,2]. Kenya has the fifth highest total number of people living with HIV with a prevalence of 4% among the adult population [3,4]. HIV is a leading cause of adult morbidity and mortality in the region [5–7]. In Kenya, HIV prevalence varies by county and Kisumu County is among the top 5 high-burden counties with a prevalence of 17.5%, which is more than four times the national average [8,9]. Kisumu County is one of 47 counties in Kenya and is further subdivided into seven sub-counties. It is in the western part of the country bordering Lake Victoria, the largest lake in Africa.

Development of novel HIV interventions requires monitoring epidemic spread, evaluating HIV prevention and treatment strategies, and assessing changes in characteristics of populations affected by HIV, particularly if demographic or geographic data may inform the HIV response [10,11]. The annual incidence of HIV among adults aged 15–64 years in Kenya is approximately 0.14% and an estimated 36,000 youths aged 15–24 years acquire HIV each year. Youth account for 42% of new HIV cases in people aged 15 years and older in the country [12–14].

Factors associated with HIV acquisition include multiple sexual partners, condomless sex, transactional sex, low socio-economic status, and alcohol and substance use. Though these factors have been well-documented globally, particularly among younger populations, there remains a need to explore specific drivers of the HIV epidemic and most vulnerable populations in specific areas with a high burden of HIV to aid in the formulation of targeted interventions [15–18]. Studies conducted in sub-Saharan Africa on factors associated with HIV acquisition indicate that risk varies widely depending on sociodemographic and behavioral factors, such as sharing needles or having condomless sex. The key risk factors for heterosexual transmission (the dominant mode of acquiring HIV in Africa) are transactional or paid sex, multiple sexual partners, co-infection with sexually transmitted infections (STIs), and lack of male circumcision [19]. Some exposures to HIV carry a much higher risk of transmission than other exposures, but risks can accumulate over time and even

relatively low-risk events can contribute to a high cumulative lifetime risk of HIV acquisition. Repeated exposures increase the chances of HIV acquisition compared to a single exposure [20,21].

Certain populations are recognized as having particularly high HIV incidence—including men who have sex with men (MSM), female sex workers (FSWs), and the fisherfolk community—who constitute a small proportion of the general population, but are at an elevated risk of HIV acquisition due to behavioral factors and social marginalization [22–25]. For example, MSM have 26 times higher risk of acquiring HIV than the general population [26]. Additionally, fisherfolk in this region have been shown to experience high mobility, transactional sexual networks, and elevated HIV prevalence, making them an important population in local HIV epidemiology [27]. Understanding HIV incidence among vulnerable groups is critical for efficient targeting of strategies to prevent HIV transmission and acquisition [28–31].

Low socioeconomic status (SES) and its correlates (lower education, poverty, and poor health) characterize many populations in low and middle-income countries such as Kenya. According to the World Bank, Kenya has one of the lowest gross domestic products in the world, with an income of $2,206.10 per capita as of 2024 [32]. Prior studies indicate that SES is differentially associated with HIV acquisition risk. HIV is intertwined in social and economic conditions, as it disproportionately affects those of lower SES. A lack of economic resources is linked to the sexual behaviors that confer HIV risk such as transactional sex [33–38].

Although aggregate incidence estimates are understood relatively well, defining HIV risk by specific locations is required to effectively conduct research of novel prevention strategies among populations most in need of these intervention. This study aimed to define HIV incidence in a cohort in Kisumu County, Western Kenya, and to identify specific factors associated with HIV acquisition. The results can aid in developing strategic, targeted, and effective HIV prevention interventions. Incidence data will also inform selection of potential sites for clinical trials to evaluate HIV vaccines and other prevention products.

## Methods

### Study Population and Procedures

Between 27 January 2017 and 17 May 2018, we enrolled participants into a non-randomized, observational, prospective study to assess HIV incidence among adults with multiple sexual partners in Kisumu, Kenya [39,40]. Males and females were eligible for enrollment if they were aged 18–35 years, tested negative for HIV, and reported sex with at least two partners in the preceding three months. A small group of people with HIV were also enrolled who satisfied other eligibility criteria in order to mask the HIV status of participants due to HIV-related stigma in the community and to serve as controls for laboratory assays. Participants in the masking group were aware of their HIV status, provided informed consent, received appropriate counseling and linkage to care, and understood that they were being enrolled as controls for laboratory assays. Exclusion criteria included pregnancy, any significant conditions (medical, psychological/psychiatric, or social) that would interfere with study conduct, intent not to reside in Kisumu County for the full study duration, and previous receipt investigational agents that could change HIV risk, such as monoclonal antibodies or candidate HIV vaccines. Determinations regarding significant underlying conditions were made by a principal or sub-investigator based on clinical judgment following review of each participant's medical and psychosocial history, rather than a predefined list of conditions. Participants were recruited via community outreach and mobilization activities facilitated by community-based organizations, local ministry of health offices, and community leaders. Targeted recruitment was conducted at locations known to be focal points for activities associated with HIV risk, including bars, markets, fishing villages, and healthcare settings such as HIV testing centers and family planning clinics.

An enrollment target of 600 participants without HIV was set based on a hypothesized HIV incidence of 3 cases per 100 person-years. Assuming retention of 85% of participants, 0 cases would need to be observed over 925 person-years in order to conclude with 95% confidence that the true HIV incidence was no less than hypothesized (95% confidence

interval 3.1-5.9 cases per 100 person-years). Participants who were lost to follow-up while the study was still open for accrual could be replaced at investigator discretion.

After enrollment, follow-up visits were scheduled every 3 months for up to 24 months. Medical history-taking, physical examination, and HIV counseling and testing were performed at each study visit. Behavioral questionnaires were administered to study participants by trained study staff at enrollment and every 6 months. The questionnaires collected information on basic socio-demographic information, general health, sexual activities, and other behaviors. Follow-up of participants was completed on 03 August 2021.

## Laboratory procedures

At study entry and quarterly visits, fingerstick whole blood was tested for HIV according to the Kenyan Ministry of Health guidelines of serial HIV rapid tests using Determine (Abbott Laboratories, Japan) and Fast Response 1-2.0 (Premier Medical Corporation, India) [41]. A third test was used to resolve inconclusive or inconsistent results, such as the HIV-1/2 rapid test or the GenScreen ULTRA HIV Ag-Ab Combo ELISA (Bio-Rad Laboratories, USA). Reactive ELISA underwent HIV confirmation/differentiation (Geenius; Bio-Rad Laboratories, USA) and/or HIV RNA PCR. All testing was performed according to package inserts. Participants with incident HIV were referred for antiretroviral therapy as clinically indicated.

## Ethics statement

This study was reviewed and approved by the Kenya Medical Research Institute Scientific and Ethical Review Unit, Nairobi, Kenya and the Walter Reed Army Institute of Research Institutional Review Board, Silver Spring, Maryland USA. The protocol used for this study complied with International Conference on Harmonization Good Clinical Practice guidelines and was conducted in accordance with the principles described in the Nuremberg Code and the Belmont Report including all federal regulations regarding the protection of human participants as described in 32 CFR 219 and Army Regulation 70–25. All participants and impartial witnesses of illiterate participants provided written informed consent before screening for study eligibility.

## Statistical analyses

Participants included in these analyses had a negative HIV test at enrollment and completed at least one follow-up visit with HIV diagnostic testing. All enrolled participants without HIV who completed at least one follow-up HIV test contributed person-time until their last documented negative test, after which they were censored. Observation time was calculated by subtracting the first date with an HIV diagnostic test result from the last date with an HIV diagnostic test result. Behavioral data were collected every six months and, if missing at a particular visit, were carried forward from the preceding visit. If HIV acquisition was detected between visits with behavioral data collection, participants were asked to complete the behavioral questionnaire at the visit when HIV was diagnosed or a separately scheduled visit soon thereafter to minimize participant burden. In the present analyses, if the interval between the visits at which HIV was diagnosed and the behavioral questionnaire was completed was longer than 10 days, data from the previous completed questionnaire closest to the HIV diagnosis date was imputed forwards to better capture behaviors prior to HIV acquisition.

Demographic and distributional response data from the screening visit were examined for differences between participants who did and did not acquire HIV using chi-squared tests of association. Subsequent analyses incorporated both static and time-variant data. Static variables were collected at screening/enrollment and included: sex, age, highest education obtained, monthly income, marital status, history of alcohol abuse, history of STI diagnosis and treatment, and age at sexual debut.

Variables allowed to be time variant included: having a current spouse or primary sexual partner, having a current secondary sexual partner, condom use with primary partner in the last 3 months, condom use with secondary partner in the

last 3 months, number of sexual partners in the last 3 months, having a partner older by 10 + years in the last 3 months, occurrences of binge drinking (defined as the consumption of 5 + alcoholic drinks in 1 day), unwanted sexual events (defined as forced sexual activity, including rape, assault, or coercion) and participation in transactional sex within the last 3 months.

HIV incidence rates were calculated overall was disaggregated by year of follow-up in the study. Per-year incidence calculation included participants with at least one valid timepoint collected in that year segment. Person-time was calculated as time between enrollment (in days) and the midpoint between the last visit with a negative HIV test and the visit with a positive HIV test. Due to a clinical pause from the COVID-19 pandemic, some participants had person-time exceeding the originally planned 2 years (or 730 days) of follow-up because their final visit was delayed. Person-time was calculated using the observation days up to the final HIV test, including the COVID-19-pause window. As there were no HIV acquisitions detected during this time, analyses of potential differences between the planned window and extended window were not conducted. Incidence rate confidence limits were calculated using Byar's method to account for the relatively low case count.

Bivariable Cox proportional hazard models were used to assess differential association between variables of interest and HIV acquisition over time. Variables that were significant in the unadjusted bivariable model were included in an adjusted multivariable model except for participant sex, which was forced selected into the multivariate model. Kaplan-Meier cumulative event curves were constructed for variables with $P \leq 0.1$ in the bivariable regression models.

Statistical analyses were conducted using SAS v9.4 (SAS Institute, Cary, NC). Kaplan-Meier curves were constructed in R (v4.3.1; R Core Team 2023).

## Results

### Summary characteristics of study participants

From 1072 participants screened for study eligibility, 671 participants were enrolled, including 619 participants without HIV (Fig 1).

Of the 619 participants without HIV enrolled, 278 (44.9%) were females (Table 1). Enrolled participants had a median age of 21 years (interquartile range [IQR]: 21–28), and the majority were single (65.9%) and had completed primary school or less education (54.1%). The median monthly income was 9000 Kenyan shillings (IQR: 5000–15000). Eleven participants (1.8%) acquired HIV during the 24-month follow-up period, of whom the majority were female (72.7%), aged 30 years and older (36.4%), single or widowed (45.5%), and had completed primary school or less education (72.7%). Most of the participants who acquired HIV (six, 54.5%) reported exchanging sex for money, goods, gifts, or favors. Although all the participants who acquired HIV reported no history of alcohol abuse, six (54.5%) sometimes consumed alcohol before having sex. None of the participants who acquired HIV had a prior history of STI diagnosis or treatment.

### HIV incidence dynamics

Table 2 depicts the incidence dynamics in the cohort. A total of 11 incident HIV cases occurred over 1117.9 person-years (PY) resulting in an incidence rate (IR) of 9.84 per 1000 PY (95% confidence interval (CI): 5.22-17.03). Of the 11 participants who acquired HIV, the majority [8] occurred during the first year of follow-up (IR: 13.84 per 1000 PY, 95% CI: 6.53-26.11). Year 2 included the COVID-19 pause (from March to September 2020), however, there were no new incident HIV cases diagnosed after this pause.

### Risk factors for HIV acquisition

In unadjusted analyses, factors associated with HIV seroconversion included number of sexual partners and forced participation in sexual activity (Table 3). Histories of STI and alcohol abuse were excluded from the bivariable results due

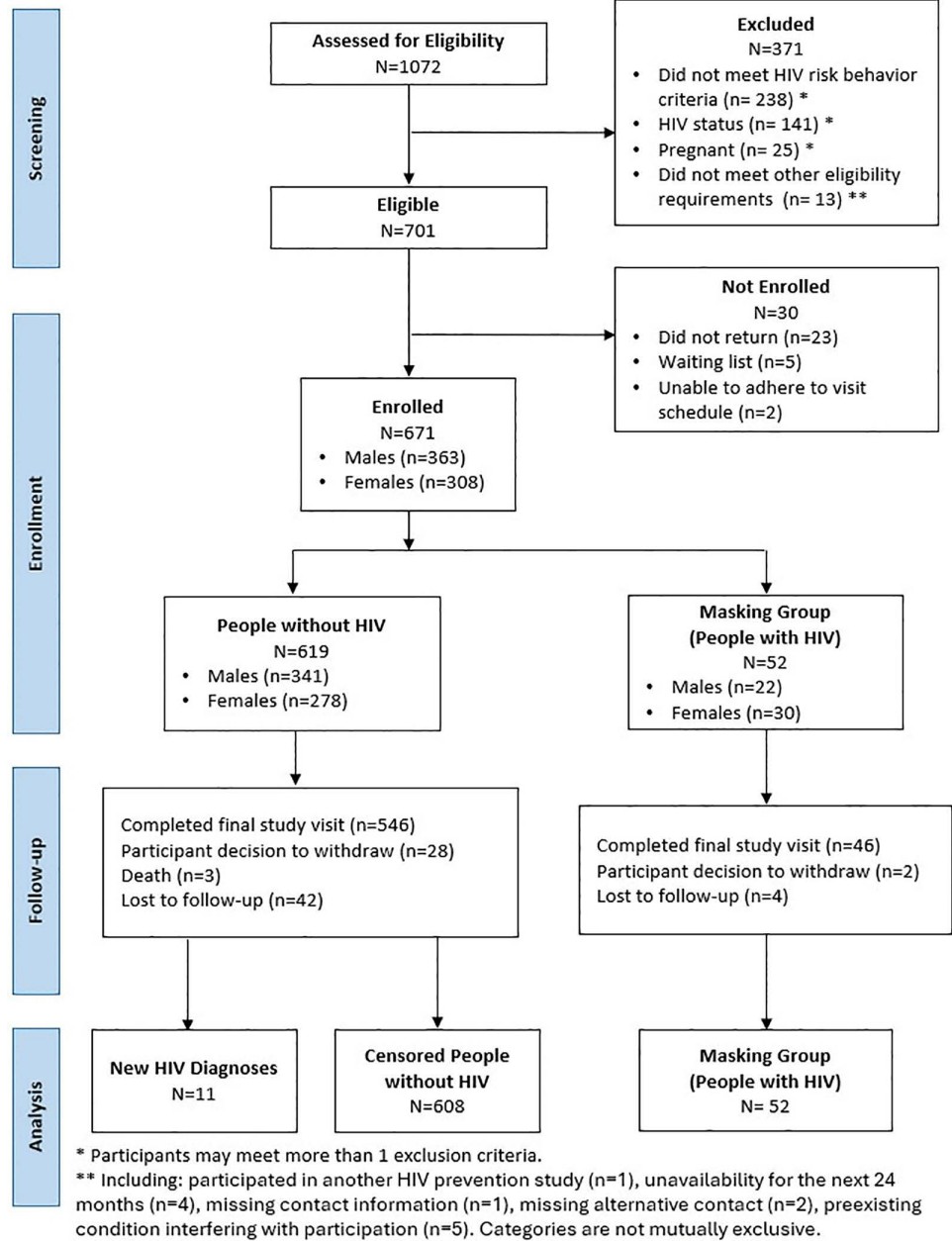

**Fig 1. RV393 Screening, Enrollment, and Follow-up Flowchart.** Participants who withdrew, were lost to follow-up, or died were included through their last HIV-testing visit.

to major violations of the proportional hazards assumption, yielding unreliable hazard ratios (HR). Three variables were included in the multivariable Cox regression analysis: sex, number of sexual partners in the last 3 months, and forced to engage in unwanted sexual events. In the multivariable analysis, females appeared to have almost 3 times greater risk of HIV acquisition compared to males, however, this association was not statistically significant (HR: 2.97; 95% CI: 0.76-11.53). Additionally, having 4 or more sexual partners and being forced to participate in sexual activity were associated with trends toward greater risk of HIV acquisition but not statistically significant after controlling for other variables.

**Table 1. Enrollment Sociodemographic and Behavioral Characteristics, by HIV Status at End of Study.**

| Characteristic | Total (N = 619) | Incident HIV case | | P |
|---|---|---|---|---|
| | | No (n = 608) | Yes (n = 11) | |
| | n (%) | n (%) | n (%) | |
| Sex | | | | 0.0613 |
| Male | 341 (55.1) | 338 (55.6) | 3 (27.3) | |
| Female | 278 (44.9) | 270 (44.4) | 8 (72.7) | |
| Age (years) | | | | 0.3655 |
| 18-20 | 110 (17.8) | 108 (17.8) | 2 (18.2) | |
| 21-24 | 223 (36.0) | 221 (36.3) | 2 (18.2) | |
| 25-29 | 176 (28.4) | 173 (28.5) | 3 (27.3) | |
| 30-35 | 110 (17.8) | 106 (17.4) | 4 (36.4) | |
| Marital status | | | | 0.1888 |
| Married (monogamous, polygamous) | 141 (22.8) | 138 (22.7) | 3 (27.3) | |
| Separated or divorced | 70 (11.3) | 67 (11.0) | 3 (27.3) | |
| Single or widowed | 408 (65.9) | 403 (66.3) | 5 (45.5) | |
| Education level | | | | 0.2114 |
| Completed primary or less | 335 (54.1) | 327 (53.8) | 8 (72.7) | |
| Some secondary or more | 284 (45.9) | 281 (46.2) | 3 (27.3) | |
| Monthly income (Ksh) | | | | 0.7199 |
| ≤9,000 | 313 (50.8) | 308 (50.9) | 5 (45.5) | |
| >9,000 | 303 (49.2) | 297 (49.1) | 6 (54.5) | |
| Missing | 3 | 3 | 0 | |
| History of sexually transmitted infection | | | | 0.5739 |
| No | 602 (97.3) | 591 (97.2) | 11 (100) | |
| Yes | 17 (2.7) | 17 (2.8) | 0 (0) | |
| History of alcohol abuse | | | | 0.3212 |
| No | 569 (91.9) | 558 (91.8) | 11(100.0) | |
| Yes | 50 (8.1) | 50 (8.2) | 0(0.0) | |
| Occurrences of ≥5 drinks in a day in the last 3 months | | | | 0.1142 |
| 0 | 516 (89.7) | 509 (90.1) | 7 (70) | |
| 1-2 | 18 (3.1) | 17 (3) | 1 (10) | |
| 3-4 | 41 (7.1) | 39(6.9) | 2 (20) | |
| Missing | 44 | 43 | 1 | |
| Drinking before sex | | | | 0.3418 |
| Never | 357 (63.3) | 352 (63.7) | 5 (45.5) | |
| Sometimes | 194 (34.4) | 188 (34) | 6 (54.5) | |
| Always | 13 (2.3) | 13 (2.4) | 0 (0) | |
| Missing | 55 | 55 | 0 | |
| History of non-injectable drug use | | | | 0.2682 |
| No | 524 (84.7) | 516 (84.9) | 8 (72.7) | |
| Yes | 95 (15.3) | 92 (15.1) | 3 (27.3) | |
| Number of sexual partners in the last 3 months | | | | 0.1090 |
| 0-1 | 293 (47.7) | 289 (47.9) | 4 (36.4) | |
| 2-3 | 233 (37.9) | 230 (38.1) | 3 (27.3) | |
| ≥4 | 88 (14.3) | 84 (13.9) | 4 (36.4) | |
| Missing | 5 | 5 | 0 | |

*(Continued)*

| Characteristic | Total (N=619) | Incident HIV case | | P |
| --- | --- | --- | --- | --- |
| | | No (n=608) | Yes (n=11) | |
| | n (%) | n (%) | n (%) | |
| Partner sex | | | | 0.3745 |
| N/A – not sexually active | 50 (8.2) | 50 (8.3) | 0 (0) | |
| Male | 247 (40.3) | 240 (39.9) | 7 (63.6) | |
| Female | 297 (48.5) | 293 (48.7) | 4 (36.4) | |
| Both | 19 (3.1) | 19 (3.2) | 0 (0) | |
| Missing | 6 | 6 | 0 | |
| Partner type: spouse | | | | 0.8431 |
| No | 373 (60.7) | 366 (60.7) | 7 (63.6) | |
| Yes | 241 (39.3) | 237 (39.3) | 4 (36.4) | |
| Missing | 5 | 5 | 0 | |
| Partner type: girlfriend/boyfriend | | | | 0.3239 |
| No | 196 (31.9) | 194 (32.2) | 2 (18.2) | |
| Yes | 418 (68.1) | 409 (67.8) | 9 (81.8) | |
| Missing | 5 | 5 | 0 | |
| Partner type: single-night encounter | | | | 0.1856 |
| No | 530 (86.3) | 522 (86.6) | 8 (72.7) | |
| Yes | 84 (13.7) | 81 (13.4) | 3 (27.3) | |
| Missing | 5 | 5 | 0 | |
| Partner type: sex worker | | | | 0.2714 |
| No | 594 (96.7) | 584 (96.8) | 10 (90.9) | |
| Yes | 20 (3.3) | 19 (3.2) | 1 (9.1) | |
| Missing | 5 | 5 | 0 | |
| Partner type: coworker | | | | 0.7006 |
| No | 606 (98.7) | 595 (98.7) | 11 (100) | |
| Yes | 8 (1.3) | 8 (1.3) | 0 (0) | |
| Missing | 5 | 5 | 0 | |
| Partner type: relative | | | | 0.8269 |
| No | 517 (84.2) | 508 (84.2) | 9 (81.8) | |
| Yes | 97 (15.8) | 95 (15.8) | 2 (18.2) | |
| Missing | 5 | 5 | 0 | |
| Partner type: other | | | | 0.8925 |
| No | 613 (99.8) | 602 (99.8) | 11 (100) | |
| Yes | 1 (0.2) | 1 (0.2) | 0 (0) | |
| Missing | 5 | 5 | 0 | |
| Currently has primary sexual partner | | | | 0.8537 |
| No | 323 (57.3) | 317 (57.3) | 6 (54.5) | |
| Yes | 241 (42.7) | 236 (42.7) | 5 (45.5) | |
| Missing | 55 | 55 | 0 | |
| Condom use with primary sexual partner | | | | 0.2309 |
| Never | 198 (81.5) | 194 (81.9) | 4 (66.7) | |
| Sometimes | 29 (11.9) | 27 (11.4) | 2 (33.3) | |
| Always | 16 (6.6) | 16 (6.8) | 0 (0) | |
| Missing | 376 | 371 | 5 | |

*(Continued)*

**Table 1.** (Continued)

| Characteristic | Total (N = 619) | Incident HIV case | | P |
| --- | --- | --- | --- | --- |
| | | No (n = 608) | Yes (n = 11) | |
| | n (%) | n (%) | n (%) | |
| Secondary sexual partner | | | | 0.4637 |
| No | 172 (28) | 170 (28.2) | 2 (18.2) | |
| Yes | 442 (72) | 433 (71.8) | 9 (81.8) | |
| Missing | 5 | 5 | 0 | |
| Condom use with secondary sexual partner | | | | **0.0279** |
| Never | 41 (9.3) | 39 (9) | 2 (22.2) | |
| Sometimes | 160 (36.3) | 154 (35.6) | 6 (66.7) | |
| Always | 240 (54.4) | 239 (55.3) | 1 (11.1) | |
| Missing / No secondary sexual partner | 178 | 176 | 2 | |
| Forced to engage in unwanted sexual events | | | | **0.0485** |
| No | 604 (98.4) | 594 (98.5) | 10 (90.9) | |
| Yes | 10 (1.6) | 9 (1.5) | 1 (9.1) | |
| Missing | 5 | 5 | 0 | |
| Exchanged sex for goods or money | | | | 0.3225 |
| No | 338 (59.9) | 333 (60.2) | 5 (45.5) | |
| Yes | 226 (40.1) | 220 (39.8) | 6 (54.5) | |
| Missing | 55 | 55 | 0 | |
| Partner younger than participant by ≥10 years | | | | 0.8136 |
| No | 544 (96.5) | 533 (96.4) | 11 (100) | |
| Yes | 19 (3.4) | 19 (3.4) | 0 (0) | |
| Refused | 1 (0.2) | 1 (0.2) | 0 (0) | |
| Missing | 55 | 55 | 0 | |
| Partner older than participant by ≥10 years | | | | 0.7084 |
| No | 435 (77.1) | 426 (77) | 9 (81.8) | |
| Yes | 129 (22.9) | 127 (23) | 2 (18.2) | |
| Missing | 55 | 55 | 0 | |

Demographic and other characteristics ascertained at enrollment were examined for differences between participants who did and did not acquire HIV using Pearson's Chi-square test or, in cases with small cell sizes, Fisher's exact test. All data are presented as n (%). Statistically significant P-values (P ≤ 0.05) are in **bold**.

Ksh, Kenyan shillings; NA, non-applicable; STI, sexually transmitted infection.

**Table 2. Incidence rates of HIV acquisition by study year.**

| | Participants who acquired HIV, n | Participants who did not acquire HIV, n | HIV incidence, rate per 1000 person-years | 95% Confidence Interval |
| --- | --- | --- | --- | --- |
| Total | 11 | 608 | 9.84 | 5.22-17.03 |
| Year 1 | 8 | 611 | 13.84 | 6.53-26.11 |
| Year 2 | 3 | 555 | 5.59 | 1.55-14.91 |

Person-time was calculated as time between enrolment (in days) and the midpoint between the last visit with a negative HIV test and the visit with a positive HIV test. Incidence rate confidence limits were calculated using Byar's method to account for the relatively low case count.

**Table 3. Unadjusted and Adjusted Cox proportional hazard models for factors associated with incident HIV.**

| Characteristic | Unadjusted Hazard Ratio (95% CI) | P | Adjusted Hazard Ratio (95% CI) | P |
|---|---|---|---|---|
| Sex | | | | |
| Male | Reference | | Reference | |
| Female | 3.44 (0.91-12.98) | 0.0679 | 2.97 (0.76-11.53) | 0.1162 |
| Age (years) | | | | |
| 18-20 | Reference | | Reference | |
| 21-24 | 0.50 (0.07-3.52) | 0.4829 | --- | |
| 25-29 | 0.94 (0.16-5.62) | 0.9446 | | |
| ≥30 | 1.99 (0.36-10.85) | 0.4280 | --- | |
| Marital status | | | | |
| Separated or divorced | 2.11 (0.43-10.44) | 0.3611 | --- | |
| Single or widowed | 0.59 (0.14-2.45) | 0.4645 | --- | |
| Education level | | | | |
| Completed primary or less | Reference | | Reference | |
| Some secondary or more | 0.44 (0.12-1.64) | 0.2207 | --- | |
| Monthly income (Ksh) | | | | |
| ≤9000 | Reference | | Reference | |
| >9000 | 1.21 (0.37-3.95) | 0.7581 | --- | |
| Occurrences of ≥5 drinks in a day in the last 3 months | | | | |
| Never | Reference | | Reference | |
| 1-2 | 4.38 (0.54-35.58) | 0.1674 | --- | |
| 3-4 | 3.56 (0.74-17.11) | 0.1136 | --- | |
| History of non-injectable drugs | | | | |
| No | Reference | | Reference | |
| Yes | 2.14 (0.57-8.06) | 0.2616 | --- | |
| Currently has primary sexual partner | | | | |
| No | Reference | | Reference | |
| Yes | 1.11 (0.34-3.64) | 0.8609 | --- | |
| Condom use with primary sexual partner | | | | |
| Never | Reference | | Reference | |
| Sometimes | 3.74 (0.69-20.43) | 0.1277 | --- | |
| Always | 0 (0−0) | 0.9953 | --- | |
| Currently has secondary sexual partner | | | | |
| No | Reference | | Reference | |
| Yes | 1.90 (0.41-8.81) | 0.4102 | | |
| Condom use with secondary sexual partner | | | | |
| Never | Reference | | Reference | |
| Sometimes | 1.87 (0.38-9.27) | 0.4429 | --- | |
| Always | 0.20 (0.02-2.18) | 0.1859 | --- | |
| Number of sexual partners in the last 3 months | | | | |
| 0-1 | Reference | | Reference | |
| 2-3 | 1.01 (0.23-4.50) | 0.9923 | 1.15 (0.25-5.24) | 0.8564 |
| ≥4 | 4.39 (1.10-17.55) | **0.0365** | 3.44 (0.84-14.15) | 0.0869 |
| Partner type: spouse | | | | |
| No | Reference | | Reference | |

*(Continued)*

**Table 3.** (Continued)

| Characteristic | Unadjusted Hazard Ratio (95% CI) | P | Adjusted Hazard Ratio (95% CI) | P |
|---|---|---|---|---|
| Yes | 0.88 (0.26-2.99) | 0.8320 | --- | |
| Partner type: girlfriend/ boyfriend | | | | |
| No | Reference | | Reference | |
| Yes | 2.20 (0.48-10.18) | 0.3132 | --- | |
| Partner type: single-night encounter | | | | |
| No | Reference | | Reference | |
| Yes | 3.08 (0.82-11.62) | 0.0965 | --- | |
| Partner type: sex worker | | | | |
| No | Reference | | Reference | |
| Yes | 3.67 (0.47-28.64) | 0.2155 | --- | |
| Forced to engage in unwanted sexual events | | | | |
| No | Reference | | Reference | |
| Yes | 9.15 (1.17-71.51) | **0.0348** | 6.16 (0.76-49.95) | 0.0886 |
| Exchanged sex for goods or money | | | | |
| No | Reference | | Reference | |
| Yes | 2.02 (0.62-6.63) | 0.2450 | --- | |
| Partner older than participant by ≥10 years | | | | |
| No | Reference | | Reference | |
| Yes | 0.88 (0.19-4.10) | 0.8756 | --- | |

Statistically significant *P*-values (*P* ≤ 0.05) are indicated in **bold** text. Sex was forced selected into the multivariable model. Variables not included from Table 1 did not meet the proportional hazards assumption.

CI, confidence interval; Ksh, Kenyan Shillings.

**Fig 2** presents the Kaplan-Meier cumulative event curves from the variables with *P* ≤ 0.1 in unadjusted models. The cumulative incidence curves for sex indicated the strongest differential incidence by sex in the first year of observation. Cumulative incidence for number of partners indicated little material difference between the 0–1 and 2–3 sexual partners in the last 3 months but indicated consistently increased incidence in participants reporting ≥4 partners. Cumulative incidence in participants reporting single night encounters was relatively homogenous in the first 6 months of the observation period but strongly differentiates after 180 days post-enrollment.

## Discussion

We observed a modest overall HIV incidence rate compared to prior studies in similar high-risk populations within sub-Saharan Africa. Similar longitudinal cohorts in high-burden regions have also reported lower-than-anticipated HIV incidence, often attributed to a cohort effect of heightened prevention behaviors resulting from study participation as well as intensified HIV prevention efforts in the community, including PEPFAR-supported interventions. For instance, a prior cohort study among serodiscordant couples in Uganda reported an incidence of 43 per 1000 person-years [42], and other incidence studies among women in South Africa had comparable results of approximately 40 per 1000 person-years [43,44]. Furthermore, a similar prospective cohort study conducted in Mozambique between 2014 and 2017 among adults aged 18–35 reported an incidence of 20.2 per 1000 person-years [45]. These comparisons suggest that the population enrolled in our study, although selected based on behavioral criteria, these measures may not have been optimal

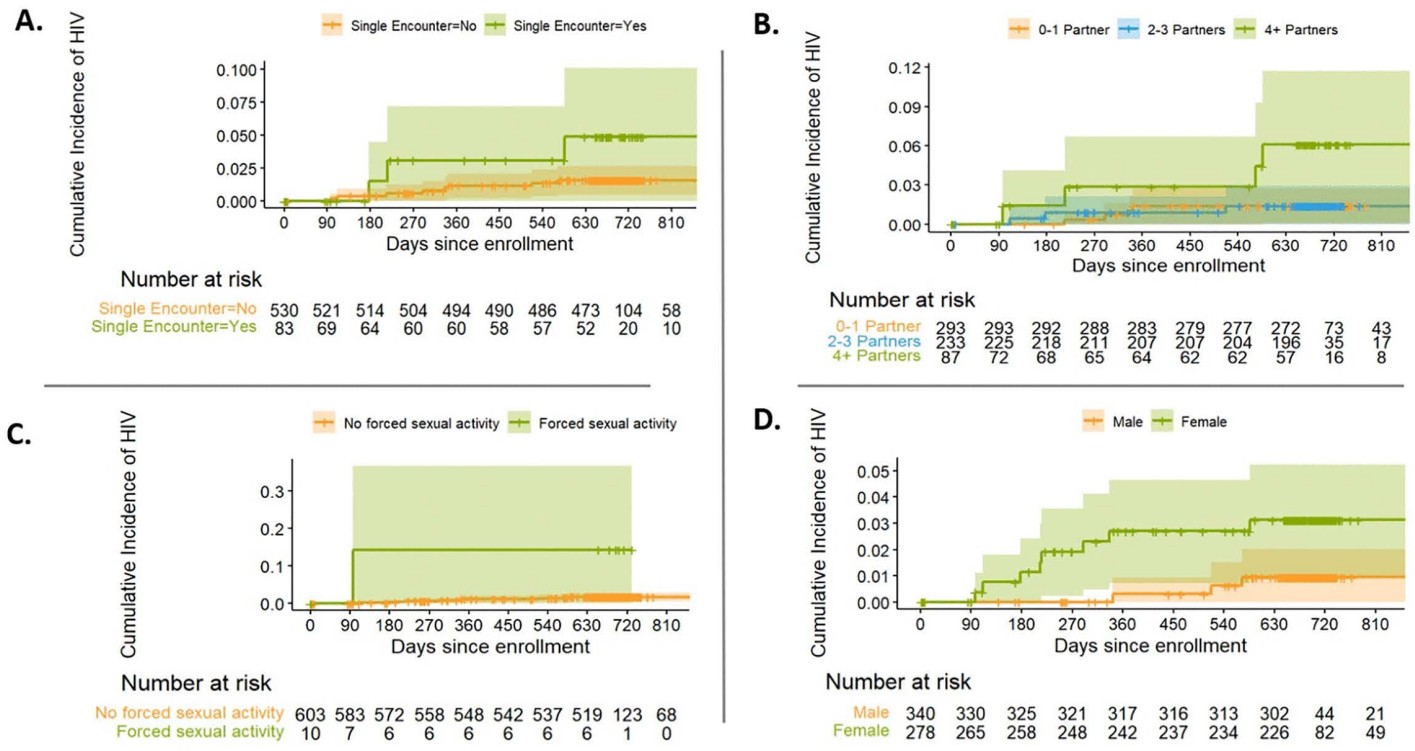

**Fig 2. Kaplan-Meier Cumulative Event Curves for Incident HIV.** Kaplan-Meier survival curves were constructed for variables with P ≤ 0.1 in the unadjusted bivariate regression models. Four covariates were examined with Kaplan-Meier curves: partner type-single night encounter **(A)**, number of sexual partners in the last 3 months **(B)**, forced to participate in sexual activity **(C)**, and sex **(D)**.

correlates of HIV risk or the enrolled population may have benefited from regional HIV prevention efforts, possibly including increased access to testing, education, and prevention services in Kenya [46].

Notably, HIV incidence in our study was highest during the first year of follow-up and declined in subsequent years. This temporal trend may reflect a combination of behavioral and external factors. Participants may have altered their risk behaviors because of repeated counseling, ongoing health education, and HIV risk reduction messaging received at study visits, a phenomenon consistent with a cohort effect seen in longitudinal studies of high-risk populations [47]. Moreover, consistent follow-up may have led to better linkage to prevention services, including condom distribution, STI screening, and HIV pre-exposure prophylaxis (PrEP). Prior analyses from this cohort have shown that almost half of participants expressed interest in PrEP, though only 18% has ever used PrEP prior to enrollment [48]. We theorize that the COVID-19 pandemic, which disrupted daily life, economic activity, and social interaction, may have also contributed to reduced sexual activity or changes in sexual networks, thereby lowering the risk of HIV acquisition in the latter study years [49]. Additionally, individuals who self-select into longitudinal studies and remain engaged in follow-up may adopt safer sexual behaviors over time, contributing to declining incidence in later study periods.

Although not statistically significant, female participants demonstrated a higher HIV risk compared to their male counterparts. Regional epidemiological data demonstrate that women (particularly adolescent girls and young women) bear a disproportionate burden of HIV in sub-Saharan Africa [5]. This sex disparity has been attributed to factors such as earlier sexual debut, higher rates of transactional sex, limited access to education and health services, and partner dynamics that limit women's ability to negotiate safer sex [50]. In our study, forced sexual activity was associated with a markedly

elevated HIV acquisition, further underscoring the vulnerability of women and the importance of addressing sexual violence as part of comprehensive HIV prevention strategies.

Our study also found that having four or more sexual partners and engaging in transactional sex were common behaviors among participants who acquired HIV. These findings align with prior research showing that multiple concurrent partnerships and transactional sex are significant drivers of HIV transmission [51]. However, none of these variables remained statistically significant in the adjusted multivariable model, likely due to the small number of HIV events, which limited statistical power. Prior analyses from this cohort have highlighted heterogeneous patterns of transactional sex (including sex in exchange primarily for money and sex in exchange for money and other necessities) that associated with other risk factors for HIV acquisition such as a higher number of sex partners [52].

There are some limitations of this study to consider. First, the relatively low number of HIV seroconversions restricted the ability to detect statistically significant associations in multivariable analyses and may have led to wide confidence intervals. Second, behavioral data such as number of sexual partners, condom use, and experience of forced sex were self-reported and thus subject to recall and social desirability bias. The 6-month interval for behavioral data collection, determined by participant-burden considerations, may have further contributed to this bias. Finally, the COVID-19 pandemic introduced unanticipated interruptions, including extended follow-up times for some participants, which may have influenced both behavior and HIV risk in ways that were not uniformly measurable. Despite these limitations, our findings have important implications for HIV prevention efforts in high-burden settings.

In conclusion, while the overall HIV incidence in this cohort was lower than anticipated, the study provides key insights into the temporal dynamics of HIV risk and the behavioral and sex-related factors that may influence transmission. Future research should aim to validate these findings in larger cohorts and assess how best to integrate HIV prevention services into community-based platforms that reach individuals before they acquire infection.

## Acknowledgments

The authors are thankful to the study participants who made this work possible. Institutions that supported this work included Kenya Medical Research Institute (KEMRI), Walter Reed Army Institute of Research-Africa (WRAIR-Africa), Henry M. Jackson Foundation for the Advancement of Military Medicine (HJF), HJF Medical Research International (HJFMRI), US Military HIV Research Program (MHRP), Walter Reed Army Institute of Research (WRAIR), and the Kenya Ministry of Health (MOH). Material has been reviewed by the Walter Reed Army Institute of Research. There is no objection to its presentation and/or publication. The opinions or assertions contained herein are the private views of the authors, and are not to be construed as official, or as reflecting true views of the Department of the Army, the Department of War, the Department of Health and Human Services, or HJF. The investigators have adhered to the policies for protection of human research participants as prescribed in AR 70−25. In addition to the masthead authors of this manuscript, the RV393 Study Group includes: Rachel Adongo (KEMRI), Rachel Aguttu (KEMRI), Hosea Akala (KEMRI), Michael Bondo (KEMRI), Erica Broach (MHRP/ HJF), Christine Busisa (KEMRI), Mark de Souza (HJF), Milicent Gogo (KEMRI), Zebiba Hassen (MHRP/ HJF), Anne Juma (KEMRI), Oscar Kasera (KEMRI), Margaret Mbuchi (KEMRI), Kayvon Modjarrad (MHRP/ HJF), Eric Ngonda (KEMRI), Jacob Nyariro (KEMRI), Jew Ochola (KEMRI), Roseline Ohore (KEMRI), Thomas Okumu (KEMRI), Mary Omondi (KEMRI), Timothy Omondi (KEMRI), Linnah Ooro (KEMRI), Beatrice Orando (KEMRI), Victorine Owira (HJF), Roselyn Oyugi (KEMRI), Eric Rono (KEMRI), Chi Tran (MHRP/ HJF), Hannah Turley (MHRP/ HJF), Clydelle Agyei (MHRP/ HJF). The lead author and Principal Investigator of the RV393 Study Group is Dr. John Owuoth (John.Owuoth@usamru-k.org).

## Author contributions

**Conceptualization:** Jessica Cowden, Merlin L. Robb, Julie A. Ake, Christina S. Polyak.

**Data curation:** Qun Li, Zebiba Hassen, Mark Milazzo.

**Formal analysis:** Adam Yates, Seth Frndak.

**Funding acquisition:** Dale J. Hu, Merlin L. Robb, Julie A. Ake.

**Investigation:** Valentine Sing'oei, June Otieno.

**Project administration:** Chiaka Nwoga, Erica Broach.

**Resources:** Michelle Imbach, Leigh Anne Eller.

**Supervision:** Nathanial K. Copeland, Christina S. Polyak, Trevor A. Crowell.

**Writing – original draft:** John Owuoth.

**Writing – review & editing:** Chiaka Nwoga, Erica Broach, Zebiba Hassen, Mark Milazzo, Tsedal Mebrahtu, Hunter J. Smith, Nathanial K. Copeland, Jessica Cowden, Dale J. Hu, Merlin L. Robb, Christina S. Polyak, Trevor A. Crowell.

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
