## [Decision Letter · Decision Letter 0]

7 Jan 2026

PONE-D-25-59538HIV Incidence Among Sexually Active Young Males and Females in Kisumu County, Western KenyaPLOS One

Dear Dr. Crowell,

Thank you for submitting your manuscript to PLOS ONE. After careful consideration, we feel that it has merit but does not fully meet PLOS ONE’s publication criteria as it currently stands. Therefore, we invite you to submit a revised version of the manuscript that addresses the points raised during the review process. Overall, I found your manuscript to be well-organized and well-written.  Please examine both reviewers' reports and provide edits and/or answers to each of their comments and questions.   

As part of this process, the journal staff requested a review of previously published RV393 data to ensure this manuscript was not a re-publication of data. I appreciate the specific methodology describing RV393's enrollment, the inclusion of Figure 1, and the disclosure statement regarding the partial presentation of this data at CROI 2025.

We look forward to receiving your revised manuscript.

Kind regards,

Darrell Eugene Singer, M.D., M.P.H.

Academic Editor

PLOS One

Journal Requirements:

3. One of the noted authors is a group or consortium [RV393 Study Group]. In addition to naming the author group, please list the individual authors and affiliations within this group in the acknowledgments section of your manuscript. Please also indicate clearly a lead author for this group along with a contact email address.

5. In the online submission form, you indicated that to request a minimal data set, please contact the Data Coordinating and Analysis Center (DCAC) at PubRequest@hivresearch.org and indicate the RV393 study along with the name of the manuscript.

6. Thank you for stating the following financial disclosure:

This work was supported by a cooperative agreement (W81XWH-11-2-0174; W81XWH-18-2-0040) between the Henry M. Jackson Foundation for the Advancement of Military Medicine, Inc., and the U.S. Department of Defense. This research was funded, in part, by the U.S. National Institute of Allergy and Infectious Diseases.

7. Please amend your list of authors on the manuscript to ensure that each author is linked to an affiliation. Authors’ affiliations should reflect the institution where the work was done (if authors moved subsequently, you can also list the new affiliation stating “current affiliation:….” as necessary).

8. One of the noted authors is a group or consortium [RV393 Study Group]. In addition to naming the author group, please list the individual authors and affiliations within this group in the acknowledgments section of your manuscript. Please also indicate clearly a lead author for this group along with a contact email address.

Reviewers' comments:

Reviewer's Responses to Questions

**Comments to the Author**

1. Is the manuscript technically sound, and do the data support the conclusions?

Reviewer #1: Yes

Reviewer #2: Yes

2. Has the statistical analysis been performed appropriately and rigorously? 

Reviewer #1: Yes

Reviewer #2: Yes

3. Have the authors made all data underlying the findings in their manuscript fully available?

Reviewer #1: Yes

Reviewer #2: No

4. Is the manuscript presented in an intelligible fashion and written in standard English?

Reviewer #1: Yes

Reviewer #2: Yes

5. Review Comments to the Author

Reviewer #1: 1. Is the manuscript technically sound, and do the data support the conclusions?

Yes, this manuscript was technically sound and supported the conclusions. Overall, well written and put together and I did not identify any large issues. A few comments/questions regarding this section, specifically for the Methods section

A) On Page 6 it is listed that exclusion criteria include “significant conditions (medial, psychological/psychiatric, or social) that would interfere with study conduct”. There are then N=24 in Figure 1 who “Did not meet other eligibility requirements”. Are these the same group of individuals? I would recommend clarifying Figure 1 to address this, potentially just with different labeling in the text box. Also, I would suggest that in the methods section that the authors try to clarify what that above statement actually means. Is Medial supposed to be “Medical”? Are there specific, objective Medical or Psychiatric conditions that the authors pre-identified that they thought would interfere with the study conduct or was each individual excluded based on best judgment by a single person or by a group of investigators? I think this part was little vague and could be written more clearly in order to best follow the methods section.

B) During the methods section and Figure 1, I was a little confused if the authors were following more of a Per-Protocol or Intention to Treat methodology for the Person-Time calculations as mentions in the statistics section on page 8 and 9 of the draft manuscript. As an example, Figure 1 lists three categories for individuals removed in follow up (decision to withdraw, death, lost to follow up). Were these individuals who were excluded included at all in the analysis if they completed questionnaires and HIV but withdrew, died, or were lost to follow up at the very end of the study (or over the COVID-19 pause mentioned in the paper)? It might be helpful to list just an additional sentence or two clarifying if any of these were used in analysis and if not, why was that decision made (could put later in the manuscript). I was also a little confused about what would happen to participants who initially did not have exclusion criteria but potentially gained exclusion criteria later in the study. As an example, if an individual was not pregnant at screening but after a time period became pregnant, how much of that was used in the analysis for Person-Time calculation? Once again, the method section was very strong and technically sound, but a few areas of clarity might help the manuscript read a little better.

2. Has the statistical analysis been performed appropriately and rigorously?

Yes, this manuscript has appropriate and rigorous statistical analysis. I only had a few comments on the statistical analysis section

A) It is very clear that authors only gathered behavioral survey data every 6 months (or up to 10 days after the HIV diagnosis) but I did not totally understand why the authors chose every 6 months versus every 3 months with the HIV testing? Was this just a logistical issue because of the burden of the survey? There is nothing wrong with this, but some clarification or justification might be beneficial. I think this is also important because the authors mention “recall and social desirability bias” on page 18, and I feel like adding length between the surveys likely added to this bias. Also, if a subject does test positive for HIV and then submits a survey, there might be a higher likelihood of recall bias if the individual already knows they have been recently diagnosed with HIV.

B) On Page 9 of the manuscript, it states “Variables that were significant in the unadjusted bivariable model were included in a fully-adjusted multivariable model except for participant sex”. I would recommend clarifying that the multivariable model adjusts for all the possible categorical variables. I think this is what was done in the statistical analysis, but it was not explicitly clear and could cause some confusion. Another option might be to edit Table 3 so the “Adjusted Relative Risk” column is a little more clear in the title or add this to the subtext you have on Page 15.

3. Have the authors made all data underlying the findings in their manuscript fully available?

Yes, I believe all the data is fully available. I had no issues with the section. As mentioned above, it might be beneficial for the authors to clarify a few things in Figure 1 and data analysis in order to improve clarity for any readers; however, I feel like all the findings and data were clearly available.

4. Is the manuscript presented in an intelligible fashion and written in standard English?

Yes, overall I felt like the manuscript is very well written, presented clearly, and that the authors did a great job with their discussion section. I had two comments regarding this section

A) On page 6 line 86 of the manuscript draft the authors include the word “medial” as an example of a significant condition for exclusion criteria. Is this word intended to be “medical”?

B) On page 9 line 150 of the manuscript the authors state that they used “Byers” method for incidence rate confidence interval calculations. Was this meant to be written as “Byar’s” method, which I believe is the recognized standard spelling? This “Byer’s” spelling is also used in Table 2 on page 13 of the draft manuscript.

Reviewer #2: Question 1: Yes

This is a well-designed prospective cohort study examining HIV incidence among adults with multiple sexual partners in a high-burden setting in Western Kenya. Overall, the study is technically sound and clearly presented. The data support the authors’ primary conclusions, which are presented with humility and are not overreaching.

The longitudinal design, frequent HIV testing using what look like pre-established algorithms, and standardized behavioral data collection are appropriate. Laboratory procedures appear rigorous, and outcome ascertainment is well explained. Ethical oversight is strong (both Kenyan and U.S. IRBs). I appreciated the explanation of informed consent procedures.

The inclusion of a masking group of participants living with HIV is explained (HIV-related stigma in the community). However, this aspect of the design raises important questions. Perhaps a brief statement clarifying that these participants were aware of their HIV status, received appropriate counseling and linkage to care, and were aware that they were being enrolled as controls would help reassure readers less familiar with this approach.

Question 2: Yes

The statistical analyses are appropriate for the study design. Incidence rates are calculated using standard person-time methods suitable for rare events, and the authors do a good job of explaining incorporation of pandemic considerations. Cox proportional hazards models are used appropriately, and violations of proportional hazards assumptions are clearly stated. I appreciated that the authors avoid overfitting and interpret adjusted analyses cautiously.

The low number of incident HIV infections is an important limitation and is acknowledged. That said, the observed incidence remains higher than national estimates cited in the Background. This raises the question of whether anticipated incidence informed enrollment targets. Power/sample size calculations are not presented, so inclusion of a brief statement describing enrollment rationale or incidence assumptions (if there were any) would help readers better contextualize the study’s power to detect associations (or even whether this was considered in the study design).

Several points of clarity could strengthen the manuscript. In the Introduction, the discussion of “assessing the feasibility of future HIV vaccine and therapeutic trials” is somewhat unclear and does not clearly flow into the Results or Discussion. This should either be more explicitly addressed later in the manuscript or removed from the abstract and introduction. My first read of the manuscript was in the “assessing the feasibility” mindset, and I do not think this was an obvious throughline. A smaller point: for those of us less familiar, the description of fisherfolk would benefit from a brief explanation of their unique characteristics, especially if they represent a large portion of the study population.

In the Methods and Results, clearer wording would improve readability. For example, line 40: the variable describing unwanted sexual events should explicitly include the term “forced.” It is used in the table, but should be used here as well. By not including the word “forced,” this variable casts a much wider net. You should be clear here what you are implying, which I think is rape, assault, or coercion. Consistent formatting of numbers throughout the Results (particularly lines 171-174) would also improve readability. Consideration could also be given to including literacy in baseline characteristics if these data were collected, though education is used and is probably getting at the same thing.

In the Discussion, several interpretations could be sharpened. Comparisons with other incidence studies (lines 220-224) would benefit from consistent use of rates per 1000 person-years. Paragraph 219-228 raises the question of whether there have been similar studies in which incidence was markedly higher, or if low incidence is a persistent limitation in these types of studies. In lines 230-233, it may also be worth noting that individuals who self-select into longitudinal studies and are able to commit to follow-up may be more likely to adopt safer sexual behaviors over time. Additionally, the statement in lines 238-239 is too strong for something not statistically significant. I recommend dropping “This trend is consistent with” (line 239) and starting the sentence with “Regional…”.

Question 3: No

The authors state that restrictions will apply. They do have appropriate statements for data availability and requests.

Question 4: Yes

Overall, this is a solid and carefully conducted study that adds useful incidence data from a high-burden setting. The results are well presented, and the discussion is humble and even-handed. The suggested revisions are largely clarifying in nature and would further strengthen the manuscript without altering its main conclusions.

6. PLOS authors have the option to publish the peer review history of their article (what does this mean?). If published, this will include your full peer review and any attached files.

Reviewer #1: No

Reviewer #2: No

---

## [Author Response · Author response to Decision Letter 1]

15 Apr 2026

RESPONSE TO EDITOR AND REVIEWER COMMENTS

Journal Requirements:

RESPONSE: Thank you, the style requirements were reviewed and the manuscript was adjusted accordingly.

RESPONSE: Thank you. The questionnaire has been completed and will be uploaded.

3. One of the noted authors is a group or consortium [RV393 Study Group]. In addition to naming the author group, please list the individual authors and affiliations within this group in the acknowledgments section of your manuscript. Please also indicate clearly a lead author for this group along with a contact email address.

RESPONSE: Thank you. The Acknowledgements have been updated as requested.

RESPONSE: Thank you – We have no restrictions to sharing a de-identified data set. Please data set can now be accessed at https://dataverse.harvard.edu/dataset.xhtml?persistentId=doi:10.7910/DVN/1KECKH . The Data Availability statement will also be revised accordingly.

5. In the online submission form, you indicated that to request a minimal data set, please contact the Data Coordinating and Analysis Center (DCAC) at PubRequest@hivresearch.org and indicate the RV393 study along with the name of the manuscript.

RESPONSE: As noted in the response above, the de-identified data set has been uploaded to a public repository.

6. Thank you for stating the following financial disclosure:

This work was supported by a cooperative agreement (W81XWH-11-2-0174; W81XWH-18-2-0040) between the Henry M. Jackson Foundation for the Advancement of Military Medicine, Inc., and the U.S. Department of Defense. This research was funded, in part, by the U.S. National Institute of Allergy and Infectious Diseases.

RESPONSE: Thank you – We confirm that the funders had no role in study design, data collection and analysis, decision to publish, or preparation of the manuscript.

7. Please amend your list of authors on the manuscript to ensure that each author is linked to an affiliation. Authors’ affiliations should reflect the institution where the work was done (if authors moved subsequently, you can also list the new affiliation stating “current affiliation:….” as necessary).

RESPONSE: Thank you – We confirm that an affiliation is linked to each author.

8. One of the noted authors is a group or consortium [RV393 Study Group]. In addition to naming the author group, please list the individual authors and affiliations within this group in the acknowledgments section of your manuscript. Please also indicate clearly a lead author for this group along with a contact email address.

RESPONSE: Thank you – The Acknowledgements have been updated as requested.

RESPONSE: Thank you – The Reviewers made no such request.

RESPONSE: Thank you, we have reviewed and updated the references to ensure they are current and appropriate. To our knowledge, no cited article has been retracted.

Reviewers' comments:

Reviewer's Responses to Questions

Comments to the Author

1. Is the manuscript technically sound, and do the data support the conclusions?

Reviewer #1: Yes

Reviewer #2: Yes

2. Has the statistical analysis been performed appropriately and rigorously?

Reviewer #1: Yes

Reviewer #2: Yes

3. Have the authors made all data underlying the findings in their manuscript fully available?

Reviewer #1: Yes

Reviewer #2: No

4. Is the manuscript presented in an intelligible fashion and written in standard English?

Reviewer #1: Yes

Reviewer #2: Yes

5. Review Comments to the Author

Reviewer #1:

1. Is the manuscript technically sound, and do the data support the conclusions?

Yes, this manuscript was technically sound and supported the conclusions. Overall, well written and put together and I did not identify any large issues. A few comments/questions regarding this section, specifically for the Methods section

A) On Page 6 it is listed that exclusion criteria include “significant conditions (medial, psychological/psychiatric, or social) that would interfere with study conduct”. There are then N=24 in Figure 1 who “Did not meet other eligibility requirements”. Are these the same group of individuals? I would recommend clarifying Figure 1 to address this, potentially just with different labeling in the text box. Also, I would suggest that in the methods section that the authors try to clarify what that above statement actually means. Is Medial supposed to be “Medical”? Are there specific, objective Medical or Psychiatric conditions that the authors pre-identified that they thought would interfere with the study conduct or was each individual excluded based on best judgment by a single person or by a group of investigators? I think this part was little vague and could be written more clearly in order to best follow the methods section.

RESPONSE: Thank you for this helpful comment. We have corrected the typographical error. Upon review, 13 participants were excluded for “other eligibility requirements,” rather than 24 as originally presented. These included participation in another HIV prevention study (n = 1), anticipated unavailability for the 24‑month follow‑up period (n = 4), missing primary contact information (n = 1), missing alternative contact information (n = 2), and pre‑existing medical or psychosocial conditions that could interfere with participation (n = 5). These categories were not mutually exclusive. We have updated the Figure 1 and have added a footer to outline the breakdown for those who “Did not meet other eligibility requirements”. Eligibility assessments were performed by a principal or sub investigator, and exclusions were based on clinical judgment, rather than a predefined list of conditions. We have revised the Methods section to clarify this process by adding the following sentence:

Determinations regarding significant underlying conditions were made by a principal or sub investigator based on clinical judgment following review of each participant’s medical and psychosocial history, rather than a predefined list of conditions (Lines 93-96).

B) During the methods section and Figure 1, I was a little confused if the authors were following more of a Per-Protocol or Intention to Treat methodology for the Person-Time calculations as mentions in the statistics section on page 8 and 9 of the draft manuscript. As an example, Figure 1 lists three categories for individuals removed in follow up (decision to withdraw, death, lost to follow up). Were these individuals who were excluded included at all in the analysis if they completed questionnaires and HIV but withdrew, died, or were lost to follow up at the very end of the study (or over the COVID-19 pause mentioned in the paper)? It might be helpful to list just an additional sentence or two clarifying if any of these were used in analysis and if not, why was that decision made (could put later in the manuscript). I was also a little confused about what would happen to participants who initially did not have exclusion criteria but potentially gained exclusion criteria later in the study. As an example, if an individual was not pregnant at screening but after a time period became pregnant, how much of that was used in the analysis for Person-Time calculation? Once again, the method section was very strong and technically sound, but a few areas of clarity might help the manuscript read a little better.

RESPONSE: Thank you for this comment. We used an ITT-style person‑time analysis in which all participants without HIV who completed at least one follow up HIV test after enrollment contributed person time until their last negative HIV test, after which they were censored. Participants who later withdrew, were lost to follow up, or died were included in the analysis for the time they were observed. Participants who developed new conditions during follow up (including pregnancy) were not removed and continued contributing person time. The COVID 19 pause did not affect incidence estimates because no seroconversions occurred during that period. We clarified our approach in the Methods section as follows:

All enrolled participants without HIV who completed at least one follow up HIV test contributed person time until their last documented negative test, after which they were censored (Lines 135-137).

2. Has the statistical analysis been performed appropriately and rigorously?

Yes, this manuscript has appropriate and rigorous statistical analysis. I only had a few comments on the statistical analysis section

A) It is very clear that authors only gathered behavioral survey data every 6 months (or up to 10 days after the HIV diagnosis) but I did not totally understand why the authors chose every 6 months versus every 3 months with the HIV testing? Was this just a logistical issue because of the burden of the survey? There is nothing wrong with this, but some clarification or justification might be beneficial. I think this is also important because the authors mention “recall and social desirability bias” on page 18, and I feel like adding length between the surveys likely added to this bias. Also, if a subject does test positive for HIV and then submits a survey, there might be a higher likelihood of recall bias if the individual already knows they have been recently diagnosed with HIV.

RESPONSE: Thank you for this comment. Behavioral questionnaires were administered every 6 months rather than every 3 months primarily due to the need to minimize participant burden within this high mobility, high risk population. We agree that a 6 month interval may increase recall and social desirability bias, as noted in the manuscript, and we have added

---

## [Editor Report · Decision Letter 1]

26 Apr 2026

HIV Incidence Among Sexually Active Young Males and Females in Kisumu County, Western Kenya

PONE-D-25-59538R1

Dear Dr. Crowell,

We’re pleased to inform you that your manuscript has been judged scientifically suitable for publication and will be formally accepted for publication once it meets all outstanding technical requirements.

Kind regards,

Darrell Eugene Singer, M.D., M.P.H.

Academic Editor

PLOS One

Additional Editor Comments (optional):

On behalf of PLOS ONE and the reviewers, I thank the authors for addressing all comments and questions with this polished revised manuscript.